# Diagnostic Mock-Up as a Surgical Reduction Guide for Crown Lengthening: Technique Description and Case Report

**DOI:** 10.3390/medicina58101360

**Published:** 2022-09-28

**Authors:** Carlos A. Jurado, Venkata Parachuru, Jose Villalobos Tinoco, Gerardo Guzman-Perez, Akimasa Tsujimoto, Ramya Javvadi, Kelvin I. Afrashtehfar

**Affiliations:** 1Woody L Hunt School of Dental Medicine, Texas Tech University Health Sciences Center El Paso, El Paso, TX 79905, USA; 2Graduate Program in Periodontics, School of Dentistry, National University of Rosario, Rosario S2000CGK, Argentina; 3Private Practice, Uriangato, Guanajuato 36260, Mexico; 4Department of Operative Dentistry, College of Dentistry and Dental Clinics, The University of Iowa, Iowa City, IA 52242, USA; 5Department of General Dentistry, School of Dentistry, Creighton University, Omaha, NE 68102, USA; 6Oasis Dental, El Paso, TX 79905, USA; 7Evidence-Based Practice Unit, Clinical Sciences Department, College of Dentistry, Ajman University, Ajman City P.O. Box 346, United Arab Emirates; 8Department of Reconstructive Dentistry and Gerodontology, School of Dental Medicine, University of Bern, CH-3010 Berne, Switzerland

**Keywords:** crowns, aesthetic dentistry, mock-up, wax-up, periodontal plastic surgery

## Abstract

*Background and Objectives:* The report describes a technique using a diagnostic mock-up as a crown-lengthening surgical guide to improve the gingival architecture. *Materials and Methods:* The patient’s primary concern was improving her smile due to her “gummy smile” and short clinical crowns. After clinical evaluation, surgical crown lengthening accompanied by maxillary central full-coverage single-unit prostheses and lateral incisor veneers was recommended. The diagnostic mock-up was placed in the patient’s maxillary anterior region and used as a soft tissue reduction guide for the gingivectomy. Once the planned gingival architecture was achieved, a flap was reflected to proceed with ostectomy in order to obtain an appropriate alveolar bone crest level using the overlay. After six months, all-ceramic crowns and porcelain veneers were provided as permanent restorations. *Results:* A diagnostic mock-up fabricated with a putty guide directly from the diagnostic wax-up can be an adequate surgical guide for crown-lengthening procedures. The diagnostic wax-up was used to fabricate the diagnostic mock-up. These results suggested that it can be used as a crown-lengthening surgical guide to modify the gingival architecture. Several advantages of the overlay used in the aesthetic complex case include: (1) providing a preview of potential restorative outcomes, (2) allowing for the appropriate positioning of gingival margins and the desired alveolar bone crest level for the crown-lengthening procedure, and (3) serving as a provisional restoration after surgery. *Conclusions:* The use of a diagnostic mock-up, which was based on a diagnostic wax-up, as the surgical guide resulted in successful crown lengthening and provisional restorations. Thus, a diagnostic overlay can be a viable option as a surgical guide for crown lengthening.

## 1. Introduction

The gingival architecture surrounding natural teeth or dental implants is an important component of aesthetics in the anterior region [1,2,3]. When the natural dentition lacks symmetry or has poor gingival architecture, these conditions can markedly alter the harmony of the dentition [4]. In recent years, it has become common for patients to have high aesthetic demands, going beyond a simple desire for a smile makeover [5,6,7,8]. Thus, clinicians must aim for an optimal gingival architecture during treatment [9]. Crown lengthening can be used in several clinical situations, such as excessive gingival display or a “gummy smile”, teeth with an inadequate amount of tooth structure, or short clinical restorations [10]. In these cases, crown lengthening can re-establish the gingival architecture and enhance the restorative outcome.

Before initiating restorative treatment with crown lengthening, the patient’s aesthetic concerns and expectations should be evaluated in detail. A diagnostic wax-up representing the desired outcome can be completed. Then, an intraoral diagnostic overlay can be fabricated to provide the patient and clinician with a tactile evaluation of the proposed treatment [11]. In addition, excellent communication between the surgeon and the restorative dentist is necessary to achieve the desired harmonious gingival architecture, especially in patients with high aesthetic demands [6,12,13,14]. Based on the diagnostic evaluations made by the restorative dentist, the surgeon can re-establish the soft and hard tissues to relocate the margins and alveolar crest and achieve periodontal health and an aesthetically pleasing gingival architecture.

Generally, a vacuum-formed surgical guide for crown lengthening is made from a duplicated cast from the diagnostic wax-up to establish the desired gingival architecture and alveolar bone crest level [15]. However, very few reports using a diagnostic overlay fabricated using a temporary bis-acrylic resin with a putty guide directly from the wax-up as a surgical guide for crown-lengthening procedures are available in the literature [16]. This case report aims to describe a technique wherein a diagnostic overlay can be used as a crown-lengthening surgical guide to help a surgeon achieve optimal gingival architecture.

## 2. Materials and Methods

A 30-year-old female patient presented to the clinic with the chief complaint of wanting to improve her smile (Figure 1). The patient had received ceramic restorations made from lithium disilicate on her central incisors two years ago. However, she disliked the results and was looking for a second opinion in an effort to improve her smile. After a detailed clinical evaluation, the patient was diagnosed with an excessive gingival display of Type IB (altered active eruption), non-ideal gingival contours, altered passive eruption of the maxillary central incisors, and incisal wear of teeth #7 and #11. She was offered the following treatment plan: (1) a crown-lengthening procedure to improve the gingival architecture, (2) replacement of the two crowns on the maxillary central incisors, (3) veneers on both lateral incisors, and (4) an incisal resin composite restoration on the left maxillary canine.

Diagnostic casts were made, and a wax-up (Wax GEO Classic, Renfert, Hilzingen, Germany) was fabricated to generate a harmonious smile according to the patient’s wishes. After showing the patient the diagnostic wax-up, a diagnostic overlay was made with temporary bis-acrylic resin (Structur Premium, VOCO, Cuxhaven, Germany). The patient consented to the treatment after approving the diagnostic wax-up and overlay. Rubber dam isolation (Nic Tone Dental Dam, MDC Dental, Guadalajara, Mexico) was placed, and the existing ceramic restorations on the central incisors were sectioned with a diamond bur (Conical End 850, Jota AG, Rüthi, Switzerland) and removed (Figure 2).

The diagnostic overlay was placed over the teeth to guide the desired contour through gingivoplasty with an electrosurgical unit (Sensimatic 700SE Electrosurge, Parkell, Edgewood, NY, USA) (Figure 3 and Figure 4).

After the new gingival architecture was achieved, buccal flap reflection provided a clear view for the surgeon performing the ostectomy. Flap reflection revealed the proper position of the osseous crest relative to the cemento-enamel junction (CEJ), which, in this case, was at the CEJ (Figure 5). An ostectomy procedure was performed using the diagnostic overlay as a guide to remove the alveolar bone. The crown-lengthening procedure was conducted within the recommended range of the Root/Crown (R/C) ratio: (1) R/C ratio of at least 1/1.5 for an abutment, and (2) R/C ratio of at least 1/1 for a crown.

The flap was repositioned (Figure 6), crown margins were refined, and provisional restorations (adjusted diagnostic overlay) were placed on the central incisors with temporary resin luting cement.

After six months, veneer preparations were performed on lateral incisors to allow for the proper healing of the periodontal complex (Figure 7), and a final impression was made with polyvinyl siloxane impression material (Virtual 380, Ivoclar Vivadent, Schaan, Liechtenstein).

The final master cast was fabricated with type IV stone (Fujirock, GC, Tokyo, Japan). Restorations, fabricated following the contours of the diagnostic wax-up, were made of refractory feldspathic porcelain (Noritake Super Porcelain EX-3, Kuraray Noritake Dental, Tokyo, Japan) for the veneers and full-coverage crowns (Figure 8).

A try-in of the final ceramic restorations was performed to evaluate the fit and contours, and the patient approved the final appearance. For bonding the ceramic restorations, isolation was provided via rubber dam placement. The teeth were air-abraded with 20-micron aluminum oxide particles (AquaCare Aluminium Oxide Air Abrasion Powder, Velopex, London, UK). The teeth receiving veneers were surface treated with 37% phosphoric acid (Total Etch, Ivoclar Vivadent, Schaan, Liechtenstein) for 15 s and then rinsed with water. A primer (Syntac Primer, Ivoclar Vivadent, Schaan, Liechtenstein ) was applied, and any excess was gently removed with air. Adhesive (Syntac Adhesive, Ivoclar Vivadent, Schaan, Liechtenstein) was applied, and any excess was removed with air according to the manufacturer’s recommendations. The intaglio surfaces of the ceramic restorations were etched with 5% hydrofluoric acid (IPS Ceramic Etching Gel, Ivoclar Vivadent, Schaan, Liechtenstein) for 60 s, and Monobond Plus (Ivoclar Vivadent, Schaan, Liechtenstein) was applied to the etched surfaces. Light-cure resin luting cement (Variolink Esthetic LC, Ivoclar Vivadent, Schaan, Liechtenstein) was applied to the veneers, and they were seated. Excess cement was removed, and the restorations were cured using an LED light-curing unit (VALO Cordless, Ultradent, South Jordan, UT, USA) on each surface (facial, palatal, mesial, and distal) for 20 s. The crowns were cemented with a dual-cure resin luting cement (Panavia V5, Kuraray Noritake Dental, Tokyo, Japan) and light-cured, followed by applying appropriate pre-treatments to the teeth and ceramic restorations.

After adjusting the occlusion as needed, the restorations were finalized with polishing points (Dialite Feather Lite, Brasseler USA Dental, Savannah, GA, USA) and polishing paste (Dialite Intra-Oral Polishing Paste, Brasseler USA Dental, Savannah, GA, USA). The incisal wear was addressed as follows. The maxillary left canine received 37% phosphoric acid etching gel for 15 s, and an adhesive (Tetric N-Bond Universal, Ivoclar Vivadent, Schaan, Liechtenstein) was applied for 20 s, gently air-thinned, and light-cured for 20 s. A nano-hybrid flowable composite resin (Tetric N-Flow, Shade A1, Ivoclar Vivadent, Schaan, Liechtenstein) was placed on the incisal edge and light-cured for 30 s. The resin composite restoration was re-shaped on the incisal edge with a fine diamond bur (Diamond bur FG 859012, Jota AG, Rüthi, Switzerland). The restoration was final-polished with green and grey composite polishers (Composite Diamond Polisher, Jota AG, Ruthi, Switzerland) using a polishing paste (Diamond Polish Mint, Ultradent, South Jordan, UT, USA) and a polishing brush (Jiffy Composite Polishing Brush, Ultradent, South Jordan, UT, USA). The patient approved of the shape and size of the final restorations, and the treatment fulfilled her aesthetic desire (Figure 9).

An occlusal night guard was also provided to prevent damage to the final restorations. At the patient’s five-year follow-up, she was fascinated with the clinical outcome (Figure 10).

## 3. Discussion

Despite the lack of clinical case reports, evidence from this case suggests that a diagnostic overlay for crown lengthening allows outcomes to be predictable for follow-up periods of at least five years. These findings are essential for the anterior region, where soft tissue changes may compromise treatment outcomes without a vacuum-formed surgical guide. Typically, the restorative process of crown lengthening using a vacuum-formed surgical guide needs both provisional restorations and a surgical guide. However, this clinical case shows that a diagnostic overlay can be used to confirm the proposed treatment plan with the patient and provide a two-for-one surgical guide and provisional restorations, thus reducing costs. Recently, a fully digital workflow for crown lengthening, using a single surgical guide, was reported [17,18,19,20]. However, this technique requires an intraoral scanner, 3D printer, and cone beam computed tomography (CBCT) scan, relying on many kinds of expertise in the digital workflow. Most clinicians are still not familiar with digital workflow [21], and the additional time and costs required before surgery are disincentives for its introduction.

In contrast, it is simpler to use conventional methods to prepare the soft and hard tissues based on a diagnostic overlay when it is placed in the mouth if the clinician knows the appropriate distances to the alveolar bone crest level from the gingival margins. Most clinicians know that the distance from the alveolar crest to the gingival margin on the facial and palatal aspects is in the range of 3 mm, while the distance from the alveolar crest to the gingival margin on the interproximal aspect is about 5 mm due to the height of the interproximal papilla [22]. Thus, it appears easier to determine the desired alveolar bone crest level using only a diagnostic overlay. This would be a simplified approach to performing crown lengthening without a traditional surgical guide.

In the present case, the use of a diagnostic overlay, based on the diagnostic wax-up, was an easy and powerful tool for the diagnostic planning of a treatment with high aesthetic demands. Given this outcome, the placement of a diagnostic overlay could be adopted as a routine protocol by clinicians, as it provides a high predictability of outcomes in aesthetically complex cases. Furthermore, the overlay can also be considered as a useful promotional tool for acquiring the patient approval of the treatment plan presented by the dental professional. In this case report, the diagnostic overlay was made based on the patient’s requests, and after it was placed in her mouth, the patient immediately indicated that she liked the result and requested the treatment.

Soft tissue crown lengthening is performed with gingivoplasty using a scalpel, an electrosurgical unit, a radiosurgical unit, or a laser [23]. If the new gingival margin position is near the underlying bone, a flap should be reflected for an ostectomy to re-establish an adequate biologic width. In the current case, the diagnostic overlay was placed and guided the use of the electrosurgical unit for the external gingivoplasty. A flap was reflected to recontour hard tissues and re-establish the biologic width of 3 mm. Compared with a traditional scalpel, an electrosurgical unit allows the clinician to cut, ablate, and re-shape soft tissues with no resulting bleeding and no need for suturing. The diagnostic overlay was an excellent guide.

Another consideration when performing crown lengthening is the healing period. The periodontal phenotype is a crucial factor, especially in aesthetic outcomes, because it impacts both the healing and final position of the gingival margin [24]. Research suggests that a thin biotype has a thickness of 1.5 mm or less, and a thick biotype has a thickness of 2.0 mm or more. Patients with a thin biotype may experience more gingival recession than those with a thick one [25]. The patient in this case had a thick biotype; thus, the likelihood of gingival recession was minimal. When considering aesthetic outcomes during this kind of treatment, the ideal healing time ranges from six weeks to six months, and a longer time may be required for patients with a thin biotype [26,27]. In the present case, the clinicians decided to wait six months before finalizing the ceramic restorations. This period provided adequate time for tissue healing and resulted in a stable gingival margin position for a pleasing aesthetic result. The diagnostic overlay was used as the provisional restoration and performed well during this time.

A limitation of this traditional workflow can happen if the diagnostic wax-up is excessive; then, it will create bulky restorations. In order to prevent this, the lip support is evaluated by the clinician and patient during the mock-up. Thus, the diagnostic overlay is necessary to demonstrate the likely outcome to the patient, and, as a provisional restoration during the healing of the hard and soft tissue, it can also be used as the surgical guide while attaining good aesthetic results. This suggests that a separate surgical guide may be unnecessary in many cases, and the assumed additional precision resulting from the use of a dedicated surgical guide may similarly be unnecessary. A simplified procedure using the diagnostic overlay may achieve all treatment goals at a lower cost.

## 4. Conclusions

The case presented in this clinical report shows that using a diagnostic overlay, which was based on a diagnostic wax-up, as the surgical guide resulted in successful crown lengthening. These results suggest that a diagnostic overlay may be viable for surgically guiding crown lengthening in aesthetically complex cases.

## Figures and Tables

**Figure 1 medicina-58-01360-f001:**
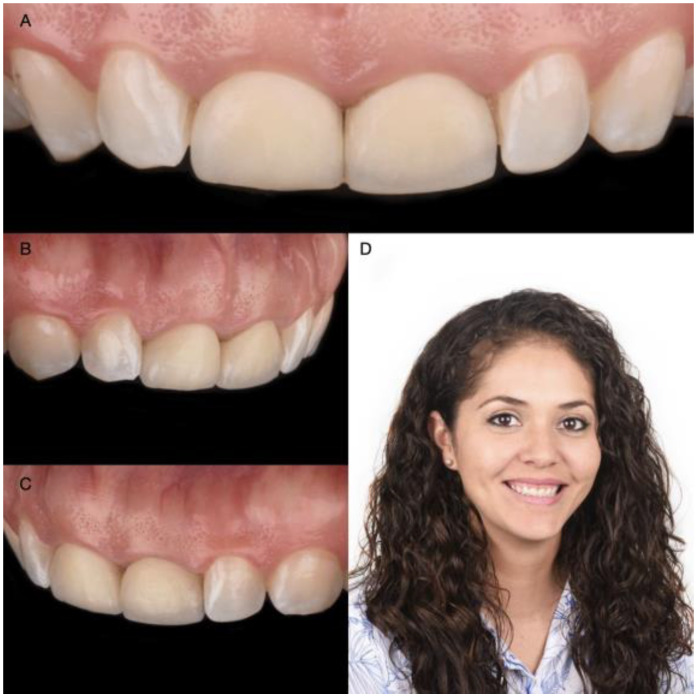
Initial scenario: (**A**) Intraoral frontal view; (**B**) Intraoral right-side view; (**C**) Intraoral left-side view; (**D**) Headshot of face smiling.

**Figure 2 medicina-58-01360-f002:**
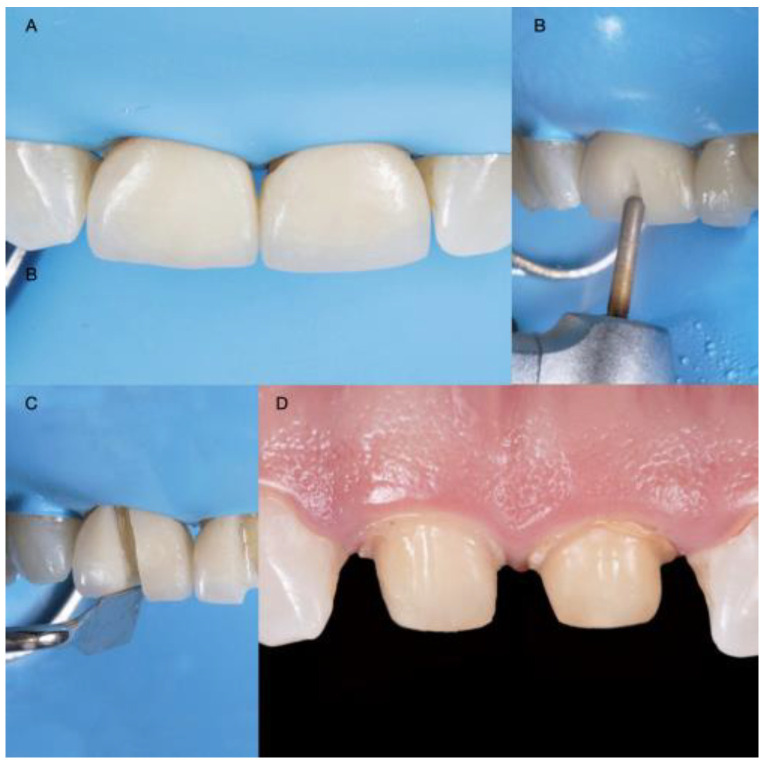
Crown prostheses removal: (**A**) Rubber dam isolation; (**B**) Initial channel in buccal surface; (**C**) Use of hand instrument for wedging; (**D**) Abutment assessment.

**Figure 3 medicina-58-01360-f003:**
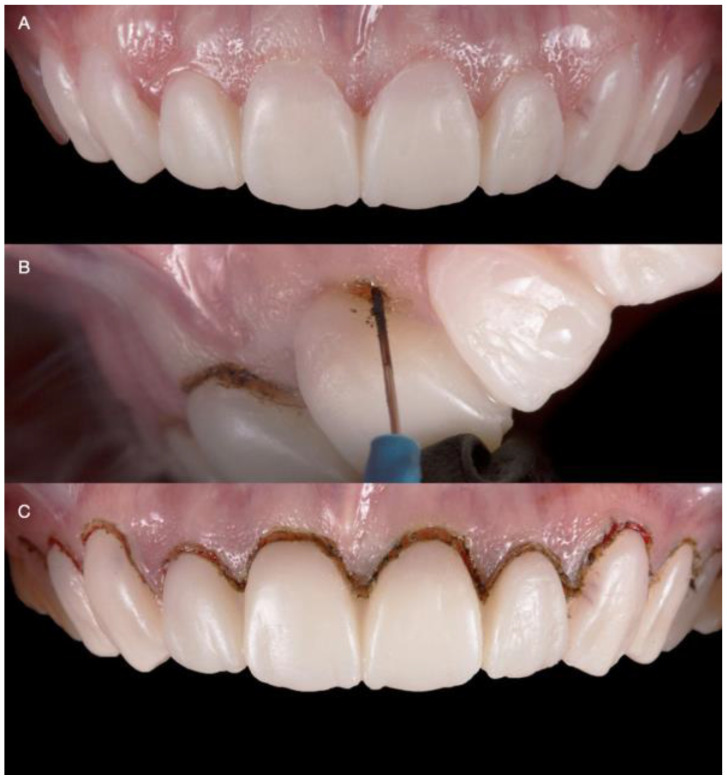
Diagnostic overlay and tissue recontour: (**A**) Placement of the diagnostic overlay intra-orally; (**B**) Gingivoplasty with an electrosurgical unit; (**C**) Finishing the contouring with the laser.

**Figure 4 medicina-58-01360-f004:**
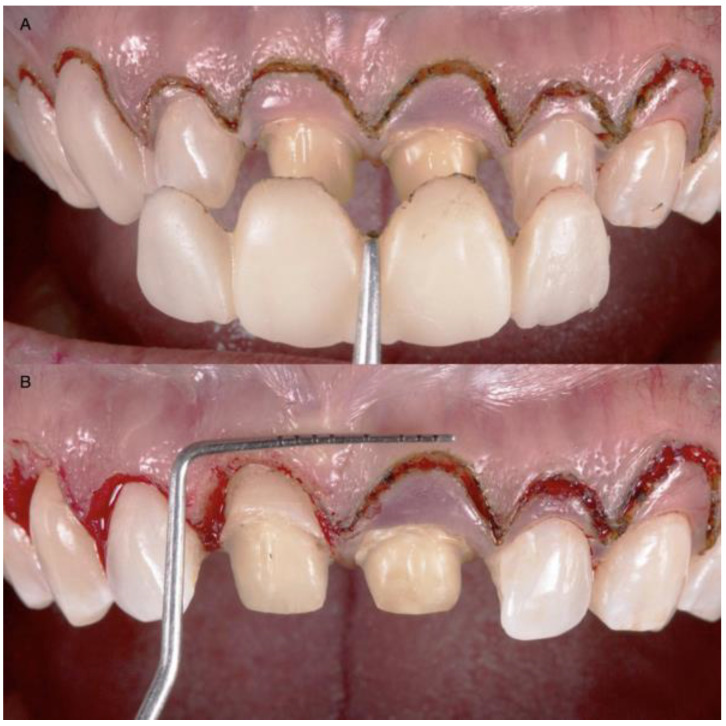
Diagnostic overlay removal and evaluation: (**A**) Removal of the diagnostic overlay; (**B**) Gingival tissue architecture evaluation.

**Figure 5 medicina-58-01360-f005:**
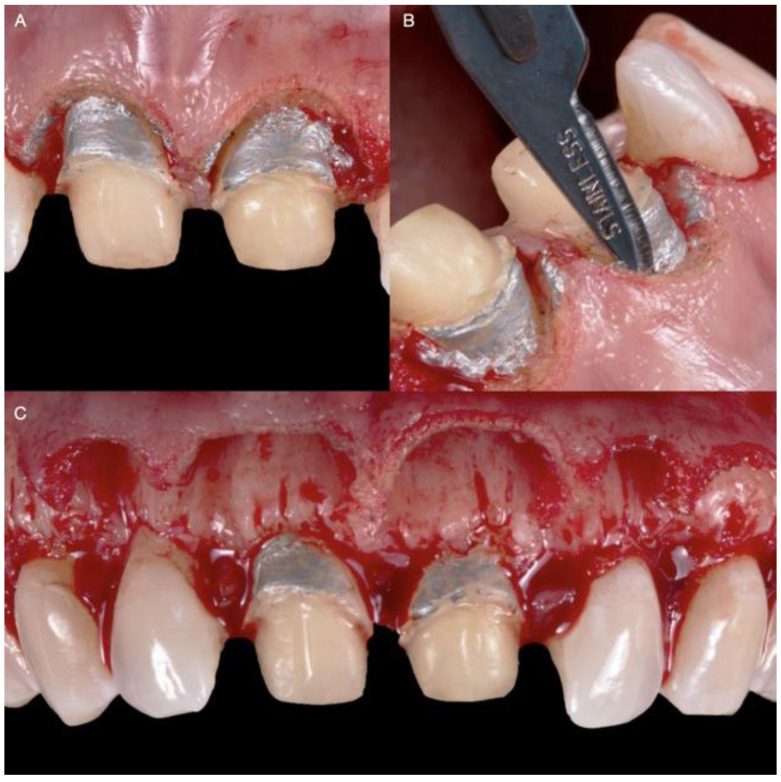
Crown-lengthening procedure: (**A**) Gingivectomy completed; (**B**) Initiation of flap; (**C**) Flap reflation.

**Figure 6 medicina-58-01360-f006:**
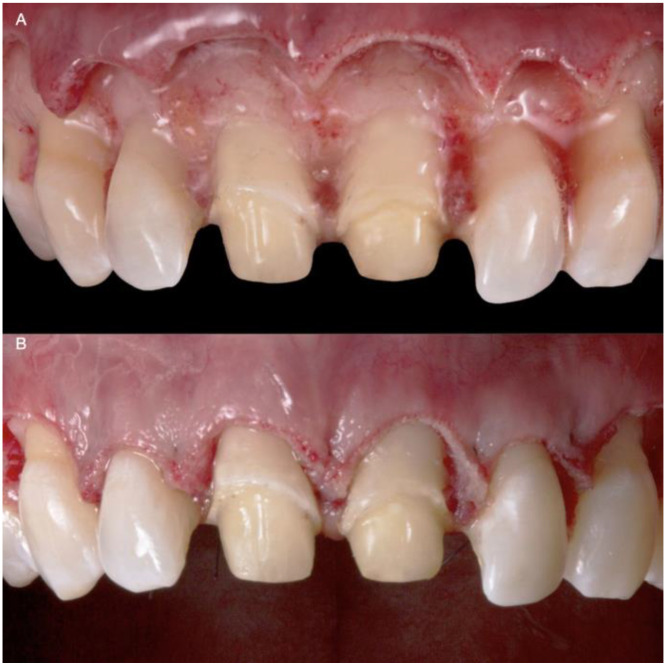
Flap release and reposition: (**A**) Flap release; (**B**) Flap reposition after suturing.

**Figure 7 medicina-58-01360-f007:**
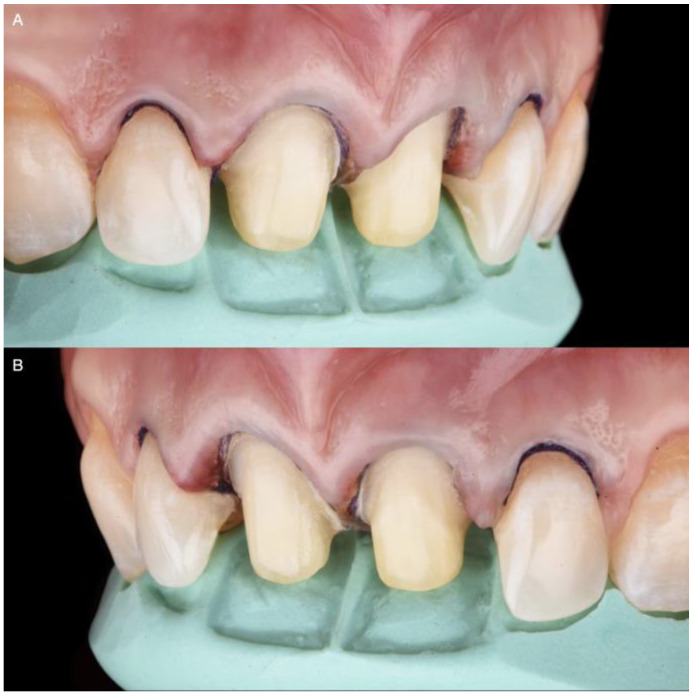
Lateral incisors veneer preparations: (**A**) Right side of the lateral veneer preparation with the reduction guide; (**B**) Left side of the lateral veneer preparation with the reduction guide.

**Figure 8 medicina-58-01360-f008:**
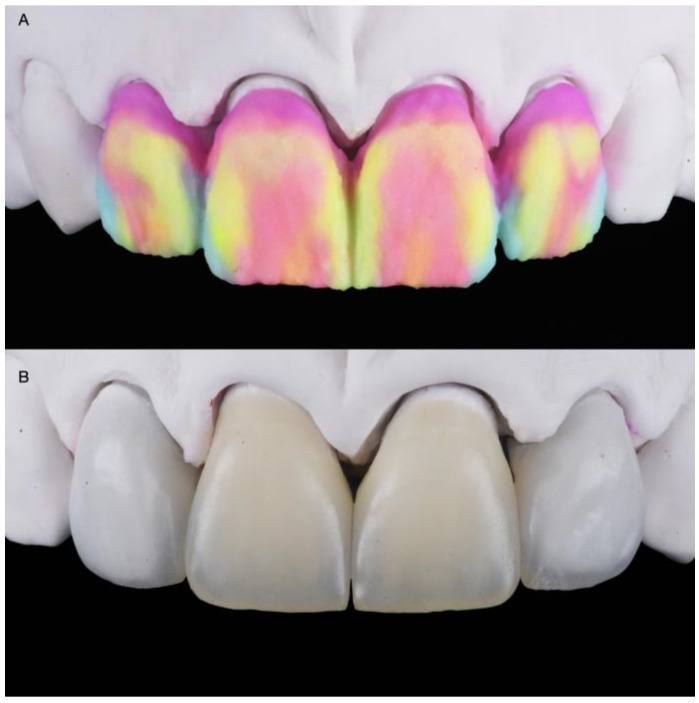
Feldspathic veneers on the master cast: (**A**) Porcelain build-up before baking; (**B**) Definitive restorations.

**Figure 9 medicina-58-01360-f009:**
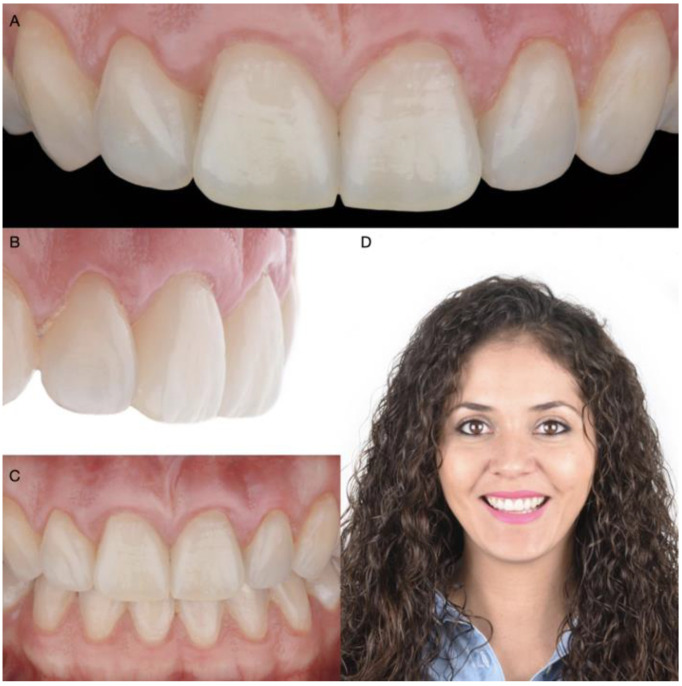
Final restorations: (**A**) Frontal view; (**B**) Lateral view; (**C**) Frontal in occlusion; (**D**) Final smile.

**Figure 10 medicina-58-01360-f010:**
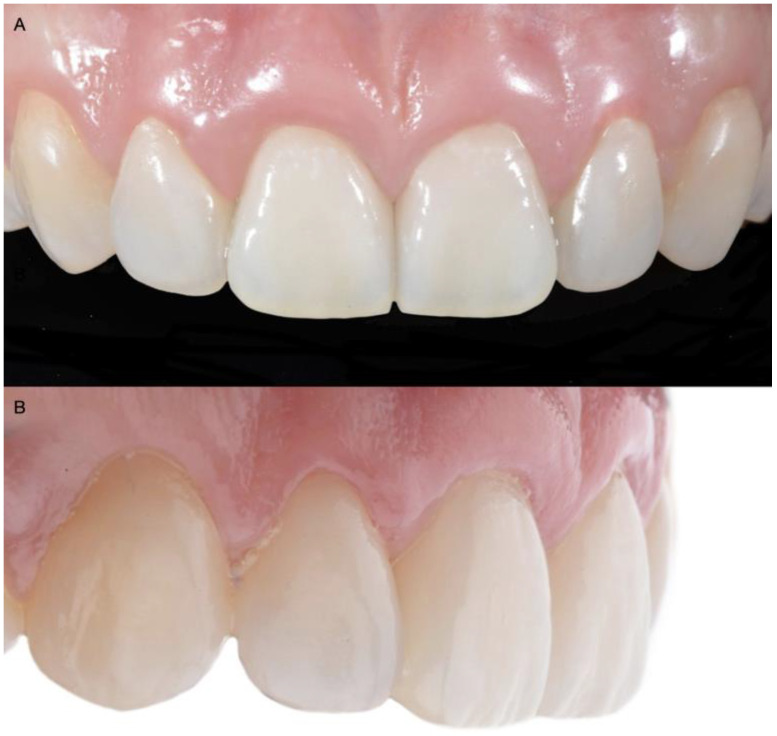
Five-year follow-up: (**A**) Frontal view; (**B**) Lateral view.

## Data Availability

Not applicable.

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
