# Peer review of "Diagnostic Mock-Up as a Surgical Reduction Guide for Crown Lengthening: Technique Description and Case Report"

_medicina, 2022, doi:10.3390/medicina58101360_

Round 1

Reviewer 1 Report

Dear authors,

Thank you for submitting this interesting case report. Although this concept might not have been published in this format before it is not necessarily a novel idea. The article is well described, execution and clinical documentation is excellent and presents a viable, realistic treatment approach. There are some aspects that need further clarification and improvement before being considered for publication. 

Title: add that it is a "case report"

Introduction:

- consider referencing the work of different groups

- using "natural teeth" instead of "dentition" in the first paragraph will read better.

- Line 45 of page 1: re-write the sentence. "removal of problems" does not seem to be appropriate

- Line 64 of page 2: review for correction if it is not an overstatement to say that there are no clinical reports using this technique. There are previous publications using at least very similar approaches. Gurrea J, Bruguera A. Wax-up and mock-up. A guide for anterior periodontal and restorative treatments. Int J Esthet Dent. 2014 Summer;9(2):146-62. PMID: 24765624.

Material and methods

- figure 1 descriptions are swapped (B) should be "right side" and (C) "left side"

- it is not clear what was the reference for bone reduction after flap was elevated. Was the diagnostic wax-up replace and used for this matter?

Discussion:

- the term osteoplasty is used in the wrong context multiple times. When bone reduction is performed for establishing a new distance for the supracrestal tissue attachment ("biological width"), ostectomy is also necessary. Please correct.

- change "periodontal biotype" for "periodontal phenotype"

- Limitations of the digital workflow were well described but limitations of such an analog workflow should also be further detailed. In order to have the restorative material over the gingival tissues to guide the soft-tissue recontour, the diagnostic wax-up needs to be overcontoured in the buccal cervical portion. This might create an idea of overcontoured, bulky teeth. Do the author see that as a limitation for patient acceptance as it might not represent exactly the contour of the "final restorations"? I believe this should be discussed.

Author Response

Reply to the reviewers

REVIEWER ONE

Dear authors,

Thank you for submitting this interesting case report. Although this concept might not have been published in this format before it is not necessarily a novel idea. The article is well described, execution and clinical documentation is excellent and presents a viable, realistic treatment approach. There are some aspects that need further clarification and improvement before being considered for publication. 

RE: Thank you, we truly appreciate your constructive review.

Title: add that it is a "case report"

RE: We updated the title by adding "a case report."

Introduction:

- consider referencing the work of different groups

RE: New references were included that reflect the support of our approach from other groups according to the referenced concepts.

- using "natural teeth" instead of "dentition" in the first paragraph will read better.

RE: The term “natural teeth” was included.

- Line 45 of page 1: re-write the sentence. "removal of problems" does not seem to be appropriate

RE:  The sentence was modified.

- Line 64 of page 2: review for correction if it is not an overstatement to say that there are no clinical reports using this technique. There are previous publications using at least very similar approaches. Gurrea J, Bruguera A. Wax-up and mock-up. A guide for anterior periodontal and restorative treatments. Int J Esthet Dent. 2014 Summer;9(2):146-62. PMID: 24765624.

RE: Thank you so much.

The new reference was included accordingly.

Material and methods

- figure 1 descriptions are swapped (B) should be "right side" and (C) "left side"

RE: The description was corrected.

Thank you for noticing.

- it is not clear what was the reference for bone reduction after flap was elevated. Was the diagnostic wax-up replace and used for this matter?

RE: Indeed, the mock-up was used to guide the bone reduction as per establishing the supracrestal tissue attachment (previously known as the biologic width) from the proposed gingival margin of the restorations (which ended up apical than the initial scenario).

Discussion:

- the term osteoplasty is used in the wrong context multiple times. When bone reduction is performed for establishing a new distance for the supracrestal tissue attachment ("biological width"), ostectomy is also necessary. Please correct.

RE: Thank you.

The term osteoplasty was replaced with ostectomy.

- change "periodontal biotype" for "periodontal phenotype"

RE: Thank you.

The term was modified.

- Limitations of the digital workflow were well described but limitations of such an analog workflow should also be further detailed. In order to have the restorative material over the gingival tissues to guide the soft-tissue recontour, the diagnostic wax-up needs to be overcontoured in the buccal cervical portion. This might create an idea of overcontoured, bulky teeth. Do the author see that as a limitation for patient acceptance as it might not represent exactly the contour of the "final restorations"? I believe this should be discussed.

RE: This limitation was included. Thank you for mentioning it.

We are delighted by the improvements to our manuscript that resulted from the reviewer’s suggestions.

Reviewer 2 Report

I congratulate with the autors for the beautiful case report/technique description.

My suggestion is to accept it in the present form but the authors should rename the article as follows: 

Diagnostic Mock-up as a Surgical Reduction Guide for Crown Lengthening: Technique description and case report.

Author Response

REVIEWER TWO

I congratulate with the authors for the beautiful case report/technique description.

RE: Thank you so much for the positive feedback.

My suggestion is to accept it in the present form but the authors should rename the article as follows: 

Diagnostic Mock-up as a Surgical Reduction Guide for Crown Lengthening: Technique description and case report.

RE: The title was modified as per the reviewer's suggestion.

We truly appreciate the time and effort invested in revising our manuscript thoroughly.
